# Fungal and Mycotoxin Contamination of Green Leaf Spices Commercialized in Romania: A Food Choice Perspective

Anca Cighir [1,2,3], Augustin Curticăpean [4,*], Anca Delia Mare [1,3], Teodora Cighir [3], Manuela Rozalia Gabor [5], Felicia Toma [1,6] and Adrian Man [1,3]

1   Department of Microbiology, George Emil Palade University of Medicine, Pharmacy, Sciences and Technology of Târgu Mureș, 38 Gheorghe Marinescu Street, 540139 Targu Mureș, Romania; anca.cighir@umfst.ro (A.C.); anca.mare@umfst.ro (A.D.M.); adrian.man@umfst.ro (A.M.)
2   Doctoral School of Medicine and Pharmacy, George Emil Palade University of Medicine, Pharmacy, Sciences and Technology of Târgu Mureș, 38 Gheorghe Marinescu Street, 540139 Targu Mureș, Romania
3   Department of Medical Laboratory, Mureș Clinical County Hospital, 1 Gheorghe Marinescu Street, 540103 Targu Mureș, Romania
4   Department of General and Inorganic Chemistry, Faculty of Pharmacy, George Emil Palade University of Medicine, Pharmacy, Sciences and Technology of Târgu Mureș, 38 Gheorghe Marinescu Street, 540139 Targu Mureș, Romania
5   Department ED1—Economic Sciences, Faculty of Economic and Law, George Emil Palade University of Medicine, Pharmacy, Science and Technology of Târgu Mureș, Livezeni Street 69, 540142 Targu Mureș, Romania; manuela.gabor@umfst.ro
6   Department of Medical Laboratory, Mureș Emergency Clinical County Hospital, 50 Gheorghe Marinescu Street, 540136 Targu Mureș, Romania
*   Correspondence: augustin.curticapean@umfst.ro

**Abstract:** A healthy, sustainable diet contributes massively to the general well-being of a population. Nowadays, people have started to significantly improve their diet by reducing processed products, as well as including a higher number of fruits, vegetables, cereals, and spices to flavor their food. However, making the right food choice, without any harmful pathogens that pose a risk to human health, can remain quite challenging. Despite prioritizing food safety in its processing, production, and distribution, food contamination remains a prevalent and undesirable occurrence. This study aims to detect the fungal load of commonly used green leaf spices commercialized in Romania and the impact of those microorganisms and their secondary metabolites on consumers. Six (28.57%) out of the twenty-one tested samples showed different degrees of fungal contamination, mostly with *Aspergillus section Flavi*, followed by *Aspergillus section Nigri* and other fungi in lower amounts. Three main fungal secondary metabolites with potential impact on consumers, namely mycotoxins, were determined with high-performance liquid chromatography (HPLC): Aflatoxin B1, Ochratoxin A, and Zearalenone. Moreover, their legal limits (5 µg/kg, 15 µg/kg, and 50 µg/kg, respectively) were exceeded by 95.24%, 100%, and 85.71%, respectively. Environmental factors that affect the processing and packaging of these spices did not show any relation to fungal contamination, conversely to price, which significantly correlates with the mycological quality of the products.

**Keywords:** health threats in food; green leaf spices; fungal load; mycotoxin

## 1. Introduction

As Thomas Edison once said, "The doctor of the future will no longer treat the human frame with drugs, but will rather cure and prevent disease with nutrition" [1]. Nowadays, one in every five deaths across the globe are attributable to a suboptimal diet, more than other risk factors, such as tobacco [2]. A multitude of chronic diseases related to an improper diet have emerged; thus, more and more experiments regarding the quality of food as a part of patient care and treatment are taking place [2].

A healthy, sustainable diet contributes massively to the general well-being of a population. As time passes, people increasingly prioritize nutritious, high-quality foods, such as fruits, vegetables, cereals, and spices, moving away from processed items. Yet, ensuring food is pathogen-free remains a challenge despite efforts to prioritize safety in processing and distribution, leading to prevalent contamination issues [3].

Spices play an important role in human history and nutrition as they contributed massively to the development of different cultures worldwide [4–6]. As an entity, spices are considered a mixture of aromatic and pungent plants that have been widely used in various cuisines since ancient times to enhance flavor, aroma, and coloring [7]. They are derived from different parts of plants (e.g., seeds, barks, stigmas, flowers, roots, leaves, etc.) and can be classified according to the part of the plant where they came from, or according to flavor [4,6]. Even more, some spices have a variety of benefits as they can be used for their antioxidant, anti-inflammatory, and antifungal effects [4,8,9].

Spices and herbs can carry bacteria, yeasts, and molds, with contamination occurring at various stages. Pre-harvesting conditions are related to climatic factors, such as high temperatures or high humidity, that make spices from the tropical area prone to a higher level of contamination [4]. Post-harvest conditions (e.g., extended drying times and improper storage, cross-contaminations due to less effective storage cleaning between yearly crops, etc.), including storage, packaging, distribution, and handling, also play crucial roles in contamination [8]. Despite the global presence of fungi in spices, measures to prevent their spread remain limited [10].

Fungi also have an important impact on immune-compromised patients, potentially causing invasive fungal infections (e.g., aspergillosis, mucormycosis) [11]. Therefore, any comorbidity or underlying condition that leads to immune suppression can lead to a high susceptibility to acquire invasive fungal infections. In healthy individuals, long-term exposure to environmental molds can cause allergic reactions [12].

Mycotoxins are natural fungal secondary metabolites with low molecular weight, which vary in chemical structure, toxicity levels, target organs, and biological effects [13]. More than 400 mycotoxins synthesized by filamentous fungi are described, and (amongst them) aflatoxins, ochratoxins, fumonisins, trichothecenes, zearalenone, and deoxynivalenol are worthy to be mentioned [12]. They are produced by different filamentous fungal species, part of *Aspergillus*, *Penicillium*, *Fusarium*, and *Alternaria* genus [12]. The toxic effects of mycotoxins in humans include immunosuppression, mutagenicity, pathogenicity, teratogenicity, and hepatotoxicity. The latter is considered to be the most dangerous, due to the increased risk of developing hepatocellular carcinoma [14–17].

This study aims to detect the fungal load of some of the most used green leaf spices commercialized in Romania, as well as the presence of three important mycotoxins (Aflatoxin B1, Ochratoxin A, and Zearalenone), all with a potential impact on human health, as described above. Furthermore, this study approaches the significance of environmental factors, such as humidity and temperature, and their impact on spice contamination, as well as the correlation between price and the fungal and mycotoxin contamination of these products. Thus, the study aims to offer a better understanding of some less-studied risk factors that can contribute to the development of several pathologies and aid the population in making better, more informed, food choices.

## 2. Materials and Methods

### 2.1. Microbiological Tests to Quantify the Fungal Load

A total number of 21 samples were tested, selected from 3 different producers, consisting of 7 green leaf spices: oregano (*Origanum vulgare*), thyme (*Thymus vulgaris*), basil (*Ocimum basilicum*), lovage (*Levisticum officinale*), parsley (*Petroselinum crispum*), dill (*Anethum graveolens*), and rosemary (*Rosmarinus officinalis*). The manufacturers were randomly chosen according to their price range: a low-priced product (A), a high-priced product (B), and an average-priced one (C). For each spice type, one product per price category was selected. The prices were noted and calculated in eurocents/g The samples, averaging around 7–10 g

per type of spice, were purchased the same day they were analyzed from a grocery store in various packages. A specific criterion was applied while selecting the samples: each spice type belonged to the same production batch. These packages contained dried and processed spices.

The fungal load was assessed by the quantitative method, Standard Plate Count [18]. For this, 5 g of each green leaf spice was weighted directly from the package into a sterile container using an analytical balance (Kern, Balingen, Germany), and then combined with 90 mL of sterile saline solution. The suspension was mixed on a magnetic steerer for 10 min at 360 rotations/minute then left for sedimentation on the table at room temperature for 30 min. From the supernatant, a 1/10 dilution was also performed in a sterile 1000 μL tube to facilitate colony counting by combining 990 μL of sterile water with 10 μL of the sample mixture. Both from the initial suspension and the 1/10 dilution, 100 μL was seeded on specific culture media (Potato Dextrose agar was selected due to its ability to grow both superior and inferior molds as well as inhibit the growth of bacteria, Czapek agar and Rose Bengal chloramphenicol agar were selected due to their capacity of selecting only superior molds, and stopping the invasive, inferior molds from growing and covering the whole media–Oxoid, UK). The suspension was added to the culture media using a pipette and then dispersed using a 10 μL sterile loop, spreading the inoculum in three directions, thus covering the whole plate surface. The media were further incubated at 32 °C (optimal temperature for mold growth) for up to 7 days to ensure proper conditions for fastidious fungi, which require a longer time to grow. The plates were checked daily for fungal growth, with the colony counting process taking place in most cases approximately 72 h after the inoculation. From each plate, the different colony morphotypes were isolated on Potato Dextrose agar for further identification.

The processing was performed in triplicate for all the samples, and the final reported number of colonies consisted of the mean number of colonies obtained on Potato Dextrose agar for each inoculation. The level of contamination was quantified using colony-forming units/gram of product (CFU/g). The value in CFU/g was calculated by taking into consideration the number of colonies, the dilution factor, and the inoculated volume, according to the following formula:

$$\frac{\text{CFU}}{\text{g}} = \frac{\text{Number of colonies} \times \text{dilution factor}}{\text{inoculated mass}}$$

Fungal identification to the level of genus was performed from pure cultures, based on the macroscopical aspect of the colony (color, granulation, growth invasiveness) and microscopical examination using lactophenol cotton blue staining (Sigma-Aldrich, St. Louis, MO, USA), looking for specific structures that could aid in identification, such as aspergillar heads and septate hyaline hyphae for *Aspergillus* spp. or nonseptate hyphae for *Rhizopus* spp. For an accurate presumptive genus identification, the macroscopical and microscopical aspects were further compared to literature data.

### 2.2. Mycotoxin Detection and Quantification Using HPLC

Besides the fungal load, mycotoxin contamination was additionally assessed using a previously validated HPLC method [18,19]. Three mycotoxins were quantified: Aflatoxin B1, Ochratoxin A, and Zearalenone. The process consisted of three different steps: the preparation of standards in methanol, mycotoxin extraction from the samples, and the identification of the extracted compounds at UV lengths of 215–362 nm (recorded at specific individual wavelengths for each compound which had different degrees of absorption).

The first step of the detection procedure consisted of preparing the samples for analysis. This involved measuring 5 g of each sample and mixing it with 10 mL of ethyl acetate, followed by stirring at a constant speed for 30 min on a magnetic stirrer. Afterward, the suspensions were centrifuged for 10 min at 4500 RPM, and 7 mL of the supernatant was transferred into 10 mL centrifuge tubes to be evaporated and then re-eluted with the mobile phase for injection into the HPLC system.

The HPLC was performed in the Dionex UltiMate 3000 system (ThermoFisher, Waltham, MA, USA) with a multichannel UV detector and an Acclaim 120 C18 column ($150 \times 4.6$ mm $\times 3$ μm). For the integration and analysis of the chromatograms, the Chromeleon 6.8 SP7 software package was used.

Results were reported for each mycotoxin in nanograms of toxin/gram, equivalent to μg/kg product, based on the calibration curve.

The mycotoxin content of the tested green leaf spices was interpreted considering the thresholds that should be respected when discussing mycotoxin quantities detected in food. According to (EU) 2023/915 Commission Regulation [20], the admitted maximum levels of mycotoxin for green leaf spices should be under 5 μg/kg for Aflatoxin B1, 15 μg/kg for Ochratoxin A, and 50 μg/kg for Zearalenone.

### 2.3. Sample Moisture Assessment

Besides the evaluation of fungal and mycotoxin burden, the physical conditions of the samples were likewise assessed. The moisture was measured by weighing one gram of spice samples in a thermobalance (Kern WPS 110S, Balingen, Germany), followed by desiccation at 130 °C until the stabilization of the product mass [18]. The difference between the two weightings represented the water content of the sample, which was then reported as a percentage with the thermobalance. All determinations were performed in triplicate.

### 2.4. Statistical Analysis

The data were recorded in spreadsheet software (Microsoft Excel 2021), anonymized, and then transferred into SPSS 23.0 (Statistical Package for the Social Sciences) to perform the statistical analysis. The Shapiro–Wilk normality test was used to study the normality of the data distribution, followed by parametric or non-parametric tests. Both categorical and continuous variables were used, as follows:

(1) All the data collected related to the fungal load and mycotoxin content were recorded as continuous variables;

(2) Categorical for

- Spice coded in SPSS with 1 for oregano, 2 for thyme, 3 for basil, 4 for lovage, 5 for parsley, 6 for dill, and 7 for rosemary;
- Producer: 1 for the low-priced spice, 2 for the high-priced spice, and 3 for the average-priced spice;
- All the mycotoxin loads with cut-off (CO) values: 1 for under CO and 2 for over CO.

Descriptive statistics were used to present and highlight the results, with several parameters being assessed: the mean $\pm$ standard deviation (min–max) for continuous variables and absolute and relative frequencies for categorical variables. Boxplots were used for graphical representations. The Pearson correlation was further applied to study the direction, strength, and statistical significance of the associations between the variables, with $p$-value < 0.05 considered statistically significant. For a more in-depth study of the different correlations between producers for variables such as price, fungal, or mycotoxin load, statistical tests such as the median test and Kruskal–Wallis with Dunn's post hoc multiple comparison were applied where deemed necessary.

The statistical methods applied in this study were used according to the specificity of the research, in which a small sample was used(<30 statistical registrations); therefore, the Kruskal–Wallis non-parametrical statistical method was applied. For Dunn's post hoc multiple comparisons, there are restrictions only referring to the number of minimum categories compared; in the current case, the three types of prices/producers were compared. Also, for Pearson correlations, there are no specific cut-off values for the number of statistical registrations.

## 3. Results

Out of the 21 samples, 6 (28.57%) tested positive for different degrees of fungal contamination (Figure 1), and all samples presented at least a small degree of mycotoxin contamination.

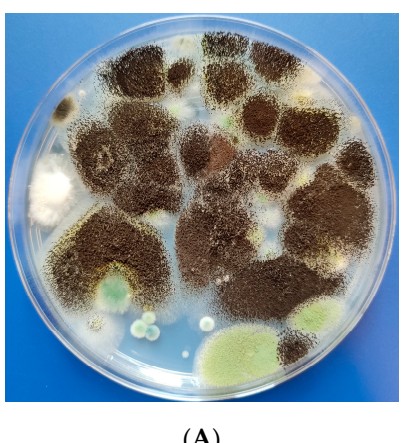
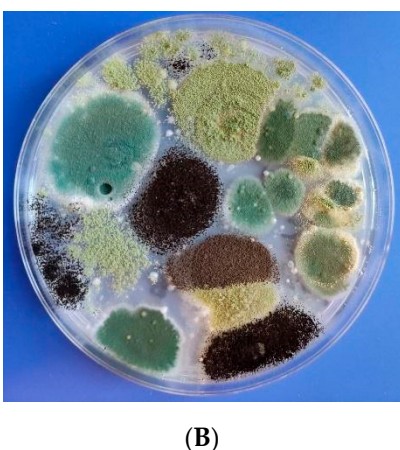
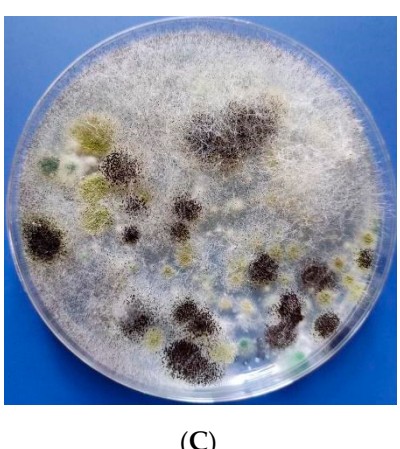

(**A**)                  (**B**)                  (**C**)

**Figure 1.** Suggested images for the plate aspect after the cultivation of different green leaf spices on Potato Dextrose agar (PDA). Plate aspect after the inoculation and incubation of (**A**) thyme suspension—characteristic colonies for different species of *Aspergillus section Flavi* and *Aspergillus section Nigri*; (**B**) parsley suspension—characteristic colonies for different species of *Aspergillus section Flavi* and *Aspergillus section Nigri*; (**C**) basil suspension—characteristic colonies for different species of *Aspergillus section Flavi, Aspergillus section Nigri,* and *Rhizopus* spp.

Table 1 comprises an overview of the samples that were contaminated with different fungal species and their fungal load based on their dry weight. The most affected sample was oregano, with an average of $(1.17 \pm 6.03) \times 10^4$ CFU/g, followed by basil, thyme, dill, and parsley. Only two spices showed no fungal contamination: lovage and rosemary. Except for basil, all samples that were contaminated originated from one single producer (33.33%), which also had the lowest price. Only one producer (33.33%) had no fungi in their samples: the producer with the highest price.

The fungal contamination was due to species from genus *Aspergillus* (*section Flavi* → n = 6; 100%; mean: $3.02 \times 10^3$ CFU/g; highest load: $5.33 \times 10^4$ CFU/g; *section Nigri* → n = 4; 66.67%; mean: $4.40 \times 10^2$ CFU/g; highest load: $4.68 \times 10^3$ CFU/g), *Absidia* (n = 1; 16.67%; mean: $8 \times 10$ CFU/g; highest load: $1.8 \times 10^3$ CFU/g), *Rhizopus* (n = 2; 33.33%; mean: $6 \times 10$ CFU/g; highest load: $1.08 \times 10^3$ CFU/g), and *Penicillium* in much lower quantities.

Regarding the fungal load based on prices, the low-priced product had the highest degree of fungal contamination, with a mean of $1.05 \times 10^4$ CFU/g, the highest fungal count was that of *Aspergillus section Flavi* ($5.33 \times 10^4$ CFU/g), while the high-priced product had the lowest degree and showed no fungal contamination. The average-priced product had a mean contamination of $3.90 \times 10^2$ CFU/g, with the most found species being *Aspergillus section Flavi* with $2.34 \times 10^3$ CFU/g (Table 2).

The water content of the samples was also considered, as higher moisture encouraged the development of fungal contaminants. The price for each spice was considered as well, since a higher price should signify a more sterile environment for packaging the spices and a higher quality of the spices, as well as better storage and transportation conditions (Table 1).

Overall, the average moisture was 9.07%, with a maximum of 14.99% (Figure 2). The product with the highest registered moisture was the average-priced product, with a maximum level of 14.99% (mean: 10.37%). The lowest moisture was found in the low-priced spice, with a maximum of 12.49% (mean: 7.62%). The high-priced spice had a maximum registered moisture of 14.56% (mean: 9.22%).

**Table 1.** The degree of fungal contamination of different green leaf spices (A—low-priced spice; B—high-priced spice; C—average-priced spice).

| Green Leaf Spice | | *Absidia* spp. | *Aspergillus section Flavi* | *Aspergillus section Nigri* | *Penicillium* spp. | *Rhizopus* spp. | Total Fungal Load | Average/Spice Type | Moisture (%) | Price (Eurocents/g) |
|---|---|---|---|---|---|---|---|---|---|---|
| | | | | | CFU/g | | | | | |
| Oregano | A | 0 | $(3.51 \pm 1.80) \times 10^4$ | 0 | 0 | 0 | $(3.51 \pm 1.80) \times 10^4$ | $(1.17 \pm 6.03) \times 10^4$ | 9.87 | 1.40 |
| | B | 0 | 0 | 0 | 0 | 0 | 0 | | 14.56 | 6.70 |
| | C | 0 | 0 | 0 | 0 | 0 | 0 | | 14.99 | 2.80 |
| Thyme | A | 0 | $1.98 \times 10^3 \pm 7.20 \times 10^2$ | $3.78 \times 10^3 \pm 9 \times 10^2$ | 0 | 0 | $(5.76 \pm 1.62) \times 10^3$ | $1.92 \times 10^3 \pm 5.40 \times 10^2$ | 10 | 1.40 |
| | B | 0 | 0 | 0 | 0 | 0 | 0 | | 13.23 | 7.10 |
| | C | 0 | 0 | 0 | 0 | 0 | 0 | | 11.66 | 2.90 |
| Basil | A | 0 | $2.25 \times 10^3 \pm 8.10 \times 10^2$ | $(1.98 \pm 1.26) \times 10^3$ | 0 | $1.80 \times 10^2$ | $(4.41 \pm 2.07) \times 10^3$ | $(7.11 \pm 2.07) \times 10^3$ | 10.74 | 1.56 |
| | B | 0 | 0 | 0 | 0 | 0 | 0 | | 10.41 | 7.10 |
| | C | 0 | $2.34 \times 10^3$ | 0 | $3.60 \times 10^2$ | 0 | $2.70 \times 10^3$ | | 11.12 | 3.00 |
| Lovage | A | 0 | 0 | 0 | 0 | 0 | 0 | No fungal contamination | 13.58 | 2.57 |
| | B | 0 | 0 | 0 | 0 | 0 | 0 | | 7.92 | 8.88 |
| | C | 0 | 0 | 0 | 0 | 0 | 0 | | 12.76 | 3.86 |
| Parsley | A | 0 | $(5.40 \pm 3.60) \times 10^2$ | $3.60 \times 10^2$ | 0 | 0 | $(9 \pm 3.60) \times 10^2$ | $(3 \pm 1.20) \times 10^2$ | 12.49 | 2.00 |
| | B | 0 | 0 | 0 | 0 | 0 | 0 | | 8.24 | 8.13 |
| | C | 0 | 0 | 0 | 0 | 0 | 0 | | 13.34 | 2.80 |
| Dill | A | $1.80 \times 10^3$ | $(9 \pm 3.60) \times 10^2$ | $9 \times 10^2$ | 0 | $1.08 \times 10^3$ | $4.68 \times 10^3 \pm 3.6 \times 10^2$ | $1.56 \times 10^3 \pm 1.20 \times 10^2$ | 9.08 | 1.75 |
| | B | 0 | 0 | 0 | 0 | 0 | 0 | | 9.22 | 8.00 |
| | C | 0 | 0 | 0 | 0 | 0 | 0 | | 10.34 | 3.50 |
| Rosemary | A | 0 | 0 | 0 | 0 | 0 | 0 | No fungal contamination | 11.19 | 1.06 |
| | B | 0 | 0 | 0 | 0 | 0 | 0 | | 11.43 | 7.10 |
| | C | 0 | 0 | 0 | 0 | 0 | 0 | | 10.04 | 3.60 |
| Average | - | $8.57 \times 10^2$ | $2.06 \times 10^3$ | $3.34 \times 10^2$ | $1.71 \times 10^2$ | $6 \times 10$ | $2.02 \times 10^3$ | $3.38 \times 10^3$ | 11.25 | 4.15 |

**Table 2.** The mean and maximum fungal load for each producer.

| Product | Variable | *Absidia* spp. | *Aspergillus section Nigri* | *Aspergillus section Flavi* | *Penicillium* spp. | *Rhizopus* spp. | Total Fungal Load/Sample |
|---|---|---|---|---|---|---|---|
| Low-priced spice (A) | Mean (CFU/g) | $2.57 \times 10^2$ | $1.31 \times 10^3$ | $8.74 \times 10^3$ | 0 | $1.80 \times 10^2$ | $1.05 \times 10^4$ |
| | Maximum (CFU/g) | $1.80 \times 10^3$ | $4.68 \times 10^3$ | $5.33 \times 10^4$ | 0 | $1.08 \times 10^3$ | $5.33 \times 10^4$ |
| High-priced spice (B) | Mean (CFU/g) | 0 | 0 | 0 | 0 | 0 | 0 |
| | Maximum (CFU/g) | 0 | 0 | 0 | 0 | 0 | 0 |
| Average-priced spice (C) | Mean (CFU/g) | 0 | 0 | $3.34 \times 10^2$ | $5.14 \times 10$ | 0 | $3.85 \times 10^2$ |
| | Maximum (CFU/g) | 0 | 0 | $2.34 \times 10^3$ | $3.60 \times 10^2$ | 0 | $2.70 \times 10^3$ |

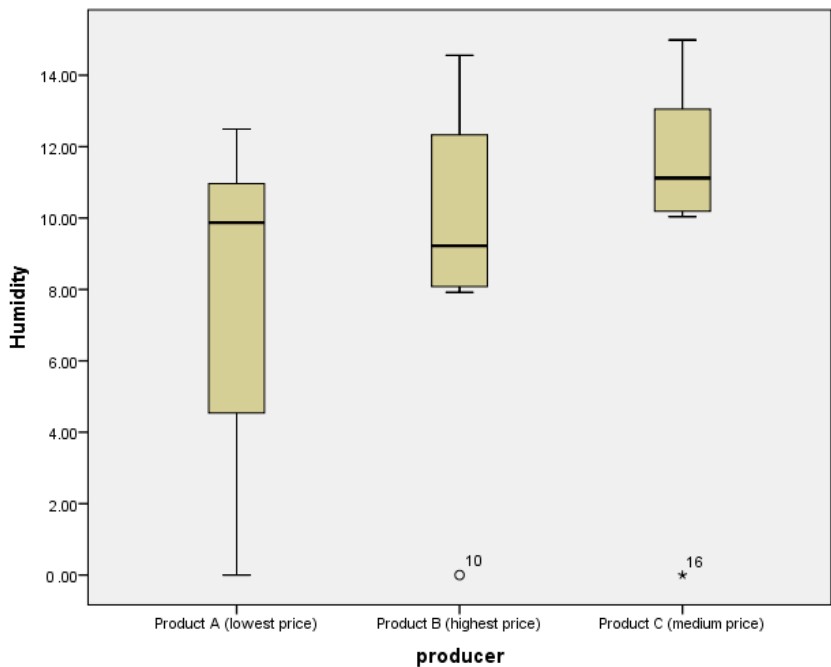

**Figure 2.** An overview of the moisture range for each sample based on products (A—low-priced spice; B—high-priced spice; C—average-priced spice). The central horizontal lines represent the median of each group. The circle (o) on the graph represents the mild outlier (values that are more extreme than the expected variation). The asterisk (*) from the graph represents an extreme outlier.

No correlation was found between the sample moisture and fungal contamination ($p = 0.5975$). On the other hand, a significant negative association was found between price and fungal contamination ($p = 0.0055$; OR = 32.5); therefore, a lower price could play a role in the increase of fungal contaminants.

Concerning price categories, one producer had the most expensive products, with prices varying between 8.88 eurocents/gram and 6.70 eurocents/gram (mean: 7.57 eurocents/gram). Another producer commercialized average-priced products, with a mean of 3.20 eurocents/gram (highest: 3.86 eurocents/gram; lowest: 2.80 eurocents/gram). Finally, the low-priced products had a mean cost of 1.67 eurocents/gram (highest: 2.57 eurocents/gram; lowest: 1.06 eurocents/gram).

The mycotoxin content was analytically determined with HPLC. Calibrations were performed to prove that the internal standard used for the simultaneous quantification, and

the three mycotoxins (Aflatoxin B1, Ochratoxin A, and Zearalenone), respect the full process of extraction from raw materials (powder from green leaf), preparation of the samples, and the final chromatographic determination (Figure S1). The lower limit of quantification LLOQ (AflaB1—2.88 µg/kg, OchraA—2.88 µg/kg, and Zeara—14.4 µg/kg) and the highest concentration values on the standards (AflaB1–288 µg/kg, OchraA—288 µg/kg and Zeara—1440 µg/kg), individually for each analyte (internal standard-phenacetin and the three selected mycotoxins), had to respect the specific rules for quantification: the regulated accepted value as concentration (in µg/kg) should be within the calibration limits, and the highest value of the calibration limit should not exceed 100 times the lower limit of quantification LLOQ. Each analyte was determined individually from the same sample at a specific wavelength, meaning that each sample injection provided four different chromatograms, and all of them should comply with a full set of chromatographic analytic requirements.

On average, all spices suffered a certain degree of contamination with at least one mycotoxin (Table 3): Aflatoxin B1 was detected over the admitted limit in 95.24% of spices (n = 20), with a mean content of $1.40 \times 10^3$ µg/kg (highest: $1.40 \times 10^4$ µg/kg; lowest: 4 µg/kg), Ochratoxin A was over the limit in 100% spices (n = 21), with a mean of $9.19 \times 10^3$ µg/kg (highest: $1.36 \times 10^5$ µg/kg; lowest: $2 \times 10$ µg/kg), and Zearalenone was detected in 85.71% (n = 18) of the products, with a mean of $6.58 \times 10^3$ µg/kg (highest: $4.02 \times 10^4$ µg/kg; lowest: 6 µg/kg)—Figure 3.

**Table 3.** Overview of mycotoxin contamination, sample moisture, and price of different green leaf spices (A—low-priced spice; B—high-priced spice; C—average-priced spice).

| Green Leaf Spice | Product | Aflatoxin B1 (µg/kg) | Ochratoxin A (µg/kg) | Zearalenone (µg/kg) | Moisture (%) | Price (Eurocents/g) |
|---|---|---|---|---|---|---|
| Oregano | A | $6.74 \times 10^2$ | $3.94 \times 10^2$ | $9.29 \times 10$ | 9.87 | 1.40 |
| | B | $1.39 \times 10^4$ | $3.20 \times 10^2$ | $1.86 \times 10^4$ | 14.56 | 6.70 |
| | C | $1.12 \times 10^4$ | $3.03 \times 10^3$ | $3.88 \times 10^4$ | 14.99 | 2.80 |
| Thyme | A | $2.28 \times 10^2$ | $2.98 \times 10^3$ | $7.06 \times 10$ | 10 | 1.40 |
| | B | $2.18 \times 10^2$ | $1.36 \times 10^5$ | $6.34 \times 10$ | 13.23 | 7.10 |
| | C | $7.04 \times 10^2$ | $2.83 \times 10^4$ | $4.02 \times 10^4$ | 11.66 | 2.90 |
| Basil | A | $4.94 \times 10^2$ | $7.14 \times 10$ | $1.76 \times 10^3$ | 10.74 | 1.56 |
| | B | $4.20 \times 10^2$ | $4.43 \times 10^3$ | $2.09 \times 10^4$ | 10.41 | 7.10 |
| | C | $9.44 \times 10^2$ | $1.28 \times 10^3$ | $1.34 \times 10^4$ | 11.12 | 3.00 |
| Lovage | A | $4.37 \times 10$ | $2.07 \times 10^2$ | $1.63 \times 10^3$ | 13.58 | 2.57 |
| | B | $1.26 \times 10^2$ | $3.12 \times 10^3$ | $7.52 \times 10^2$ | 7.92 | 8.88 |
| | C | $1.42 \times 10$ | $1.89 \times 10^3$ | $2.84 \times 10^2$ | 12.76 | 3.86 |
| Parsley | A | $1.39 \times 10^2$ | $1.99 \times 10^2$ | $3.82 \times 10$ | 12.49 | 2.00 |
| | B | $7.63 \times 10$ | $8.87 \times 10$ | 5.94 | 8.24 | 8.13 |
| | C | $3.41 \times 10$ | $2.34 \times 10^2$ | $4.04 \times 10$ | 13.34 | 2.80 |
| Dill | A | $3.35 \times 10$ | $2.37 \times 10^2$ | $8.37 \times 10^2$ | 9.08 | 1.75 |
| | B | 4.64 | $1.62 \times 10$ | $7.50 \times 10$ | 9.22 | 8.00 |
| | C | 6.93 | $3.09 \times 10^2$ | $1.31 \times 10^2$ | 10.34 | 3.50 |
| Rosemary | A | $4.61 \times 10$ | $3.43 \times 10^2$ | $2.02 \times 10^2$ | 11.19 | 1.06 |
| | B | $7.86 \times 10$ | $9.19 \times 10^3$ | $2.31 \times 10^2$ | 11.43 | 7.10 |
| | C | $2.76 \times 10$ | $7.88 \times 10$ | $1.01 \times 10^2$ | 10.04 | 3.60 |
| Average (µg/kg) | - | $5.25 \times 10^3$ | $9.19 \times 10^3$ | $8.20 \times 10^3$ | 11.25 | 4.15 |

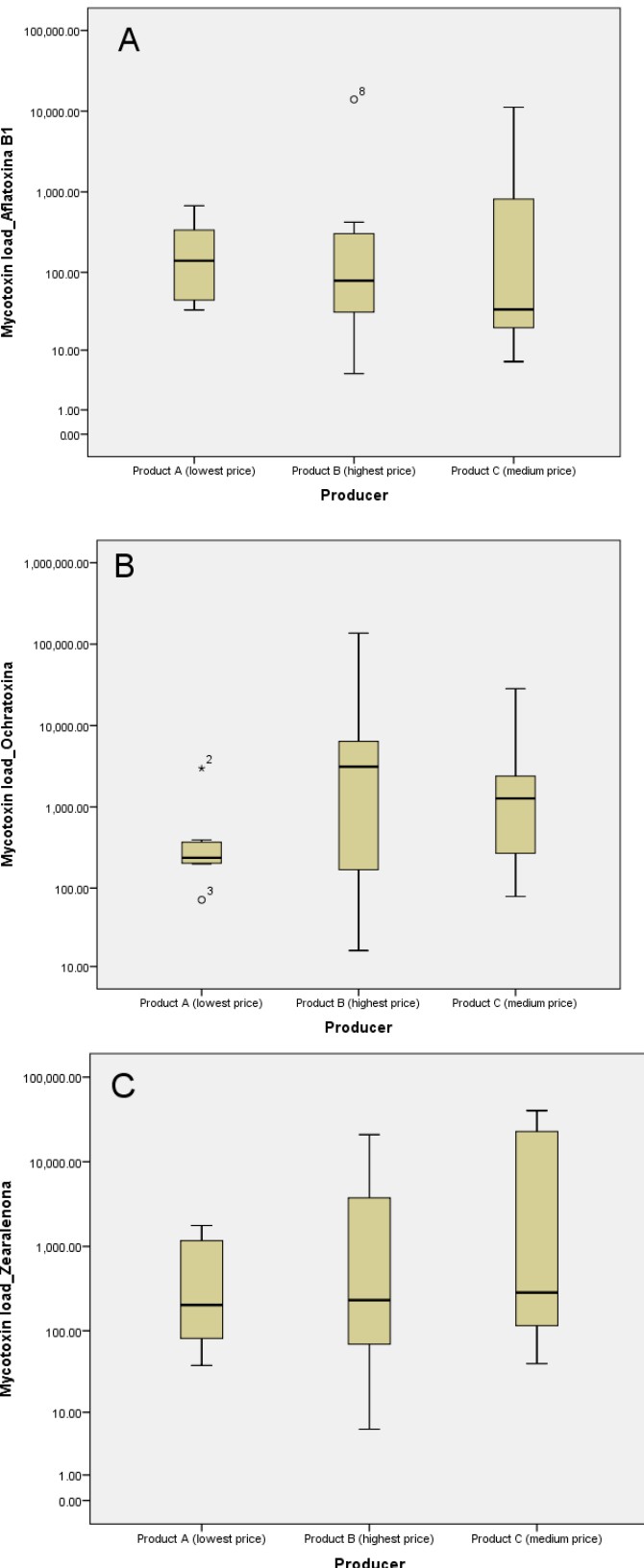

**Figure 3.** Boxplots representing the average mycotoxin content by product: (**A**) Aflatoxin B1; (**B**) Ochratoxin A; (**C**) Zearalenone. The products are labeled as A—low-priced spice; B—high-priced spice; C—average-priced spice. The central horizontal lines represent the median of each group. The circles (o) on graphs (**A**,**B**) represent the mild outlier values (values that are more extreme than the expected variation). The asterisk (*) from graph (**B**) represents the extreme outlier.

For the low-priced and the average-priced products, all samples presented Aflatoxin B1 and Ochratoxin A levels over the admitted limit for green leaf spices (over 5 µg/kg and 15 µg/kg, respectively), while Zearalenone was under the admitted limit (50 µg/kg) for one spice (14.3%), and over the admitted limit for six spices (85.7%).

For the high-priced spices, Aflatoxin B1 and Zearalenone showed levels under the admitted limit of 5 µg/kg and 50 µg/kg, respectively, for one spice (14.3%), and over the limit for six spices (85.7%); all determinations (100%) were over the admitted limit for Ochratoxin A.

The average mycotoxin contamination was also calculated for each price category. Surprisingly, although the category with the most expensive products had no fungal contamination, they contained by far the highest degree of mycotoxin contamination, with $3.40 \times 10^4$ µg/kg, which is almost double compared to the average-priced ones, possibly due to a previous contamination, but with nonviable fungal spores at the moment of this study.

Figure 4 shows representative HPLC chromatograms for a single sample containing all three mycotoxins at high levels. The figure comprises four different chromatograms recorded simultaneously for only one sample, each analyte (internal standard and the three selected mycotoxins) at specific wavelengths, as specified by the validated HPLC method [18]. The graphs emphasize the individual chromatographic peaks used for the calculations of mycotoxin concentrations from a single sample. Each chromatographic peak is labeled according to the analyte designated to be quantified, and the exact values of each recorded wavelength are mentioned in the graphs.

Pearson's correlation was also applied to find associations between the studied variables (Table S1). A strong positive direct correlation (r = +0.992; *p* = 0.000) was found between the total fungal load and *Aspergillus section Flavi*, proving that, in these specific cases, this is the main fungal component that leads to the increase in the total fungal load, followed by all the other fungal species.

For both the low-priced products (r = +0.796; *p* = 0.032) and the average-priced products (r = +0.820; *p* = 0.024), a strong positive correlation was found between the levels of Aflatoxin B1 and the total fungal load, as well as the load of *Aspergillus section Flavi*. Therefore, it could be considered that, in the present case, the fungal contamination with *Aspergillus section Flavi* is related to the Aflatoxin B1 contamination of the tested spices.

Statistical tests (median test and Kruskal–Wallis) were further applied to check if there is a statistical relationship between the previously mentioned parameters. A statistically significant difference was found between the low-priced spices and the high-priced spices and the average-priced spices, respectively, mostly in relation to the fungal load of *Aspergillus section Nigri* (*p* = 0.025 for all three); therefore, a lower price can signify a higher chance of becoming contaminated with the before-mentioned species. Regarding the fungal load of *Aspergillus section Flavi*, statistically significant differences were found only between the high- and low-priced products (*p* = 0.025), proving again that the lower the price, the higher the risk of spice contamination.

Regarding the price of the analyzed samples and its influence on their fungal contamination, a statistically significant correlation was also found between the total fungal load and the price/gram of spice (*p* = 0.013). Dunn's post hoc multiple comparison further proved that among the three producers, the one with the highest price is less likely to be contaminated than the one with the smallest price, with a statistically significant difference between low-priced spices, high-priced spices (*p* = 0.011), and average-priced spices (*p* = 0.045), respectively. Therefore, the price could have an impact on the degree of fungal contamination of the tested spices.

Concerning the mycotoxin content of all three producers, no statistical significance was found with the median test, nor with the Kruskal–Wallis test (*p* > 0.05).

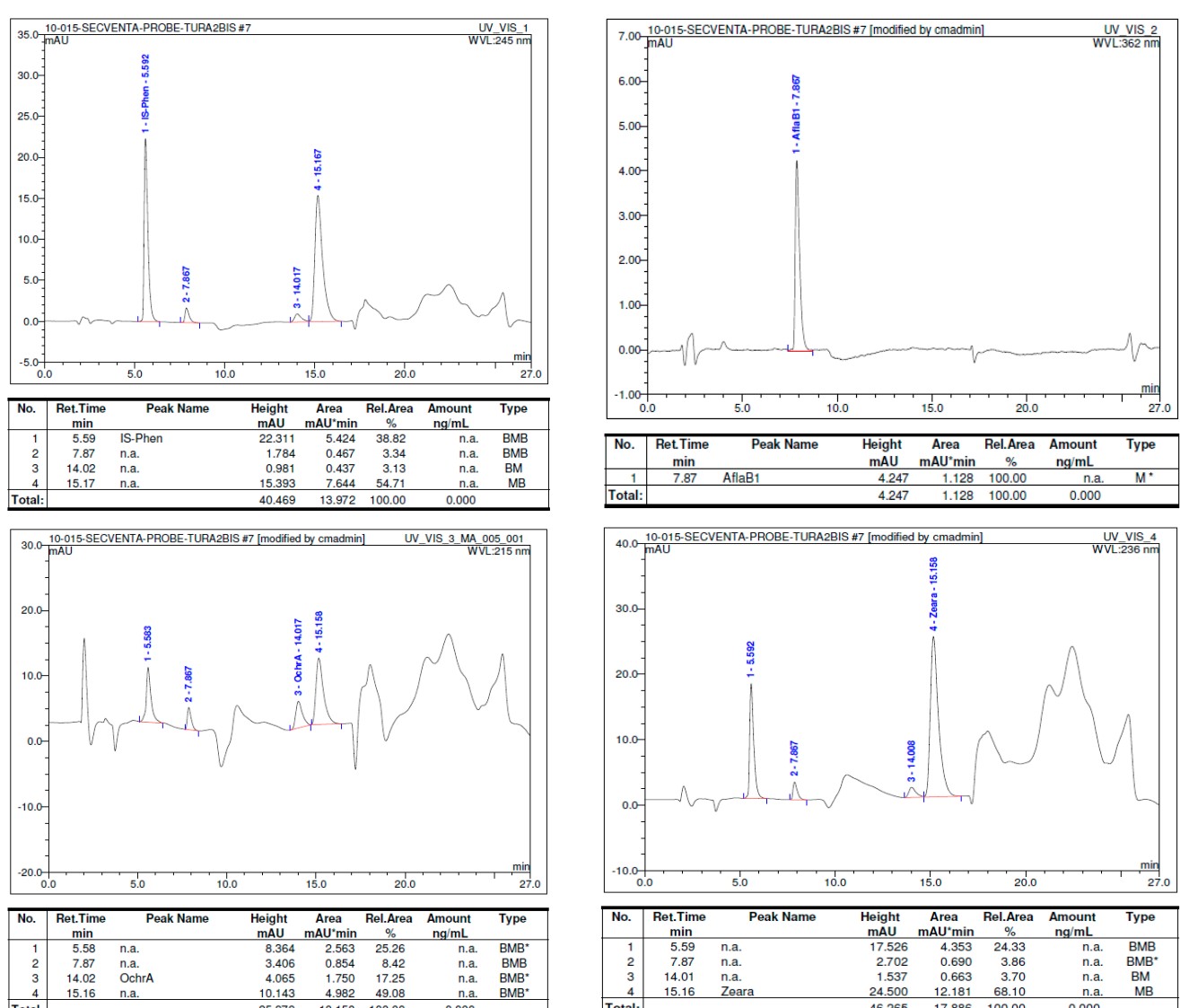

**Figure 4.** Representative HPLC chromatogram for a single highly contaminated sample. The asterisks (*) indicate that the baseline of the peak was adjusted.

## 4. Discussion

Many herbal spices, such as oregano, thyme, and rosemary, have been proven to have medicinal properties, being of strong aid in alleviating symptoms of different diseases as well as preventing others [9]. However, can these benefits still be taken into consideration when such a high degree of fungal and mycotoxin contamination is present?

Currently, very few studies regarding the contamination of food products are available, both in Romania and worldwide, and most of them are related to cereal products [18,21–24]. Thus, this research aimed to analyze the mycoflora and the mycotoxin contamination of the most common green leaf spices that are commercialized in Romania. In European countries, oregano is considered the most consumed herbal spice, followed by basil, parsley, thyme, and chives [25].

Contamination of food or agricultural commodities with different types of mycotoxigenic fungi represents a concerning issue for both human and animal health. Molds can naturally contaminate foods, as they are ubiquitous in the environment; under favorable environmental conditions, such as suitable temperature, pH, and nutrient availability, they can further multiply and produce their toxic metabolites (mycotoxins) [26].

Global mycotoxin contamination in food persists despite strict regulations on processing, storage, and agricultural practices. Mycotoxins remain unpredictable and unavoidable due to their resilience against heat and various treatments. Around 25% of crops worldwide are contaminated by mycotoxins, resulting in substantial industrial and economic losses [27]. Despite the tremendous progress that has been made in recent years, there are still major challenges in the detection of mycotoxins: difficulties in detecting low levels of mycotoxin contamination, complex food matrices that require complicated extraction processes, or the co-occurrence of mycotoxins [27].

Fungi fall into two wide categories: field and storage fungi. Field fungi are the ones that cause damage to the living plants in fields (e.g., *Alternaria* spp., *Fusarium* spp.). Storage molds are those that are mostly encountered on plant parts that are stored in high moisture conditions (e.g., *Aspergillus* spp. and *Penicillium* spp.) [28]. During their processing, spices are usually kept close to the ground which leads to moisture transfer, thus allowing the contamination as well as multiplication of potentially toxigenic fungi and also the cross-contamination between the crops [3].

Several studies show rates of fungal contamination remarkably similar to the ones determined in this study. Hashem et al. [13] studied the fungal and mycotoxin contamination of several spices found in Saudi Arabian markets, and they found a similar mycoflora, as the most common fungal contaminants were represented by species from the genus *Aspergillus*, and in lesser amounts *Penicillium*, and *Rhizopus*, but the degrees of contamination were opposite. Besides the environmental conditions, the differences may also be related to the part of the plant that was harvested; studies show that if the part of the plant where the spice was derived from is closer to the soil, the degree of humidity is higher, and therefore the chance of contamination is higher [3]. These results are also supported by Garcia et al. [4], describing similar fungal species as the most common contaminants. Our study found similarly high rates of contamination for oregano, but no contamination for rosemary. This could be due to either the way the samples are manipulated and processed in different countries or the different climatic areas that could alter the physical parameters, such as heat and humidity, which further influence the risk of contamination.

Another earlier research by Man et al. [18] regarding the fungal contamination of different spices commercialized in Romania also showed similar fungal contaminants to the present ones: *Aspergillus* spp., *Rhizopus* spp., *Penicillium* spp., and *Absidia* spp. This could be induced by the fact that, even though the spices that were studied were different from the ones in this study, the climatic area and the manufacturers were similar.

Several studies, such as those conducted by Hammami et al. [29] and Gutarowska et al. [30], reported diverse degrees of fungal contamination in herbs such as basil, oregano, dill, thyme, and rosemary. The quantities varied significantly between studies, showing inconsistencies in contamination levels among different research findings. Furthermore, the genera that were frequently identified were also similar to the ones found in this study, with a high prevalence of species from genus *Aspergillus*, *Penicillium,* and *Alternaria*. A study by the same authors [31] found higher levels of fungal contamination in fresh parsley leaves compared to this study. This difference may be attributed to the drying and manufacturing processes that parsley undergoes before reaching the shelves and being purchased by customers.

Climate is the key force that contributes to fungal colonization and mycotoxin production. Thus, any changes in environmental factors, such as temperature, humidity, or $CO_2$ concentrations, could lead to an unexpected increase in the risk of mycotoxin contamination of crops, which will further lead to huge economic losses. Global warming can also impact the probability of contamination with both fungi and mycotoxins, as all the climatic changes will affect the distribution of certain pathogens, favor the emergence of new diseases, and affect the mycoflora [32].

Mycotoxin production usually occurs as a response to different environmental conditions [12]. Worldwide, the maximum admitted limit for any type of mycotoxin and any type of food product is less than 20 µg/kg. At first, there was no limit specifically made for

spices and herbs; however, the (EU) 2023/915 Commission Regulation of 25 April 2023 [20] now sets some specific thresholds for all spices. For Aflatoxin B1, the admitted limit is under 5 μg/kg, for Ochratoxin A, under 15 μg/kg, and for Zearalenone, under 50 μg/kg.

Extended studies regarding the mycotoxin contamination of food products are available. Pickova et al. [7] conducted a review spanning five years (2015–2020), focusing on aflatoxins. Contrary to their findings where thyme rarely showed contamination and oregano/basil were uncontaminated, this study reported high concentrations of Aflatoxin B1 in all three herbs, surpassing European regulations. Ochratoxin A is the second most analyzed mycotoxin after Aflatoxins [7,33]. In most studies, oregano, cloves, thyme, and basil were typically free from Ochratoxin A contamination [7]. However, the current findings show alarmingly high levels of Ochratoxins in all these spices, exceeding established safety limits. Zearalenone is one of the least analyzed mycotoxins [7]; therefore, very little data are available about it. In this case, studies showed that the mycotoxin content proved to be low in thyme and none in basil and oregano [7]. Like with the other mycotoxins, we have found high levels of mycotoxins for all three of the above-mentioned green leaf spices. The elevated mycotoxin levels suggest inadequate handling and preservation of the spices commercialized in the Romanian market. In many spices, mycotoxin levels exceeded safety thresholds set by the European Union and other global organizations, potentially posing health risks. Prolonged consumption of such spices, especially in high quantities, may lead to the accumulation of mycotoxins in the body, heightening the risk of various associated health issues [34].

This study revealed an intriguing finding: despite having a high price, certain spices were sterile from a mycological perspective, but were highly contaminated with mycotoxins. This suggests improper manipulation, storage, and processing of spices, but proper sterilization techniques, causing destruction of the thermolabile fungi and preservation of the thermostable mycotoxins. Consequently, while appearing mycologically safe, these spices might harbor surviving fungi capable of producing toxic metabolites and thus posing health risks. Cheaper spices, likely processed under less controlled conditions, could retain viable fungi even after packaging.

It is important to note the limitations of this study, which are mainly related to the small number of processed samples. Nevertheless, the statistics showed promising outcomes, encouraging more comprehensive future research on food safety. All the statistical methods were carefully chosen for the small sample and, therefore, all the results can be inferred for entire populations with similar features.

## 5. Conclusions

The intense fungal and mycotoxin contamination of the spices that were included in this research should raise a big concern regarding the actual daily food consumption of a population, as long-term exposure can lead to negative effects on human health.

The results of this research reveal that, although there are certain limits imposed regarding the fungal and mycotoxin contamination of food, there is still a possibility of consuming products that have a high level of contamination. As mentioned in the results section, the detected fungal loads varied greatly, from no fungal contamination present in the high-priced products to a mean of $1.05 \times 10^4$ CFU/g for the low-priced products. Furthermore, the mycotoxin content was even more worrying, as all the products included in this study had at least a small degree of contamination. Moreover, even though the high-priced products showed no fungal contamination, they had a high degree of mycotoxin contamination, with a mean of $3.4 \times 10^4$ μg/kg. Aflatoxin B1, Ochratoxin A, and Zearalenone's legal limits (5 μg/kg, 15 μg/kg, and 50 μg/kg, respectively) were exceeded by 95.24%, 100%, and 85.71%, respectively, in the tested spices.

Taking this into account, it is safe to say that strict regulations regarding the allowed content of fungal and mycotoxin contamination should be enforced and respected thoroughly to ensure the future health of various populations.

**Supplementary Materials:** The following supporting information can be downloaded at: https://www.mdpi.com/article/10.3390/su152316437/s1. Figure S1. The calibration curves (HPLC standards) used for the quantification of mycotoxins from the green leaf spices samples; Table S1. Comparison between the fungal load, mycotoxin content, humidity, and price of the samples. The statistically significant *p* values are marked in bold (the threshold for the *p* values is set at 0.05).

**Author Contributions:** Conceptualization, A.C. (Anca Cighir), A.C. (Augustin Curticăpean), A.D.M., M.R.G., T.C., F.T. and A.M.; methodology, A.C. (Anca Cighir), A.C. (Augustin Curticăpean) and A.M.; validation, A.C. (Anca Cighir), A.C. (Augustin Curticăpean), A.M., F.T. and A.D.M.; formal analysis, M.R.G.; investigation, A.C. (Anca Cighir), A.C. (Augustin Curticăpean), A.D.M., M.R.G., T.C., F.T. and A.M.; writing—original draft preparation, A.C. (Anca Cighir); writing—review and editing, A.C. (Augustin Curticăpean), A.D.M., M.R.G., T.C., F.T. and A.M.; supervision, A.M. All authors have read and agreed to the published version of the manuscript.

**Funding:** This research received no external funding.

**Institutional Review Board Statement:** Not applicable.

**Informed Consent Statement:** Not applicable.

**Data Availability Statement:** Data are available upon request from the authors.

**Conflicts of Interest:** The authors declare no conflict of interest.

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
