# Peer review of "Fungal and Mycotoxin Contamination of Green Leaf Spices Commercialized in Romania: A Food Choice Perspective"

_sustainability, doi:10.3390/su152316437_

Round 1

Reviewer 1 Report

Comments and Suggestions for Authors

The work addresses the fungal detection and toxins of some of the most used and commercialized spices in Romania and the importance of environmental factors.

The experimental designs related to microbiology were well designed and substantiated.

However, the discussion about the real values of spices is not relevant as it is very regional. These values may vary from place to place and there is no consensus.

The work would be better if the leaves had the same processing for home use. For example, wash before carrying out experiments. This way the results would be more conclusive. This way, the microbial load and toxin values would change.

Author Response

Dear Reviewer,

Please find attached the cover letter containing the answers we provided to your feedback, together with the new version of the manuscript.

Kind regards,

Cighir Anca

Reviewer 2 Report

Comments and Suggestions for Authors

The manuscript "Fungal and Mycotoxin Contamination of Green Leaf Spices Commercialized in Romania: A Food Choice Perspective" requires thorough revisions, particularly in terms of English language and technical details such as spacing between numbers and units. Here are some specific considerations regarding the manuscript:

- The introduction is excessively long (1175 words) and contains numerous general references. It should be streamlined to include only the most relevant information directly related to the manuscript;

- The same applies to the discussion section, which is overly lengthy and should be condensed to better integrate and elucidate the results;

- Line 26 - "...free from microorganisms and completely healthy...": This is not accurate. Many food products are considered healthy precisely because of the microorganisms they contain;

- Line 27 - "...as food is becoming more and more susceptible to fungal contamination...": What is the basis for this assumption?

- Line 63 - "Spices have a variety of benefits and they can be used as preservatives or for their antioxidant, anti-inflammatory, and antifungal effects.": This is a generalization. Is this true for all known spices?

- Line 124 - "...as well as the influence of price on fungal and mycotoxin contamination...": This is not correct. Price itself has no effect on fungal or mycotoxin contamination. The authors likely meant to refer to a correlation between price and microbial quality;

- Line 129 - "2.1. Microbiological tests to detect the fungal load": In microbiology, detection is not the same as quantification. The authors presented results for quantification, and a certain amount is quantified, not detected;

- Line 137 - "...the common supermarket...": Correct the grammar and specify when the spices were obtained;

- Line 140 - Reference 20 for the Standard Plate Count? Is this correct? This is a self-citation. Please assess the appropriateness of this reference;

- Line 143 - "...sterile saline.": Is this referring to a sterile saline solution?;

- Line 144 - " ... then left for sedimentation on the table at room temperature.": For how long?;

- Lines 152-153 - "was then dispersed on the culture media using a sterile loop...": Is this correct? A total of 100 μl were dispersed with a loop? This seems unlikely, as loops are typically used to spread much smaller volumes (1-10 μl). Or has been used some kind of special loop?;

- Line 176 - "...followed...": Please correct the English;

- Line 205 - "...percentage by thermobalance.": Please correct the English;

- Figure 3 - Some characters are present inside images A and B, the meaning of which is not specified in the text;

- Engineer's notation is not consistently applied throughout the text. It should be expressed as a decimal fraction (greater than or equal to 1 but less than 10) followed by the appropriate power of ten;

- Line 360 - "...our samples...": Avoid using the first person (both here and throughout the text);

- The conclusions should be more concise and contain fewer general references.

Author Response

(The authors gave the same response as above.)

Reviewer 3 Report

Comments and Suggestions for Authors

Dear Authors,

In your study you examine a widespread problem of utmost importance for safeguarding human and animal health. I find that the analysis, although the number of samples examined is low, highlights a clear problem in the marketing and production of spices.

The analysis methodology is clear and straightforward, and the results are also clearly expressed.

The only thing I would suggest is to reduce the discussions as at some points the focus of the discussion is somewhat lost. In particular, I would remove what is reported from line 469 to 486 since, among the parameters that you have considered for the spice analysis (moisture and price), the sterilization method has not been considered and therefore I do not find it necessary to elaborate on that aspect which takes the focus away from your results.

Having made this minor correction I believe the article can be accepted.

Author Response

(The authors gave the same response as above.)

Reviewer 4 Report

Comments and Suggestions for Authors

This manuscript has certain research significance. However, there are the following issues that need improvement.

1. Line 27. I think the content of this sentence is not accurate.

2. Lines 37-38. This sentence described inaccurately.

3. The introduction section needs to be condensed.

4. Lines 145-146. Please change “μl” to “μL”.

5. Lines 167-173. Identifying strains also requires the use of molecular biology methods, such as identifying 16S RNA and ITS sequences.

6. Line 183. Please change “five grams” to “5 g”.

7. Lines 281-282. This sentence described inaccurately.

8. Regarding the figures, the text on the coordinate axes is too small.

9. Regarding the discussion section, it is recommended to focus on key aspects related to the research content.

10. Regarding the conclusion section, it needs to be condensed.

Author Response

(The authors gave the same response as above.)

Round 2

Reviewer 2 Report

Comments and Suggestions for Authors

In general, the authors have considered the issues raised during the first review of the manuscript and made some corrections. However, the introduction and discussion remain excessively lengthy, containing unnecessary generalities and redundancies. In the abstract, the phrase "...which could significantly correlate to the mycological quality of the products." seems to imply the possibility of a correlation between price and the microbiological quality of the products. Nevertheless, the authors have calculated statistical significances, so it should not be presented as a possibility but rather as an observed fact.

Author Response

(The authors gave the same response as above.)

Reviewer 4 Report

Comments and Suggestions for Authors

The authors had improved the manuscript. It is recommended to accept it.

Author Response

(The authors gave the same response as above.)
